# Impact of Sensorineural Hearing Loss during the Pandemic of COVID-19 on the Appearance of Depressive Symptoms, Anxiety and Stress

**DOI:** 10.3390/medicina58020233

**Published:** 2022-02-03

**Authors:** Emilija M. Zivkovic Marinkov, Natasa K. Rancic, Dusan R. Milisavljevic, Milan D. Stankovic, Vuk D. Milosevic, Marina M. Malobabic, Irena N. Popovic, Aleksandra M. Ignjatovic, Mila R. Bojanovic, Jasmina D. Stojanovic

**Affiliations:** 1Faculty of Medicine, University of Nis, 18000 Nis, Serbia; emilijazm@gmail.com (E.M.Z.M.); dusanorldusan@gmail.com (D.R.M.); milanorlstankovic@gmail.com (M.D.S.); vuk.milosevic@gmail.com (V.D.M.); drsalea@yahoo.com (A.M.I.); milabojanovic@yahoo.com (M.R.B.); 2ENT Clinic, University Clinical Center of Nis, 18000 Nis, Serbia; 3Institute for Public Health Nis, 18000 Nis, Serbia; 4Clinic for Neurology, University Clinical Center of Nis, 18000 Nis, Serbia; marinasudimac@gmail.com; 5Special Psyhiatric Hospital Gornja Toponica Nis, 18000 Nis, Serbia; ipopovic69@yahoo.com; 6Department of Otorhinolaringology, Clinical Center Kragijevac, 34 000 Kragujevac, Serbia; fonijatarkg@gmail.com; 7Faculty of Medical Science, University of Science Kragujevac, 34 000 Kragujevac, Serbia

**Keywords:** bilateral sensorineural hearing impairment, face masks, stress, anxiety, depression

## Abstract

Background. The incidence of hearing loss is constantly increasing and according to the World Health Organization, by 2050, 900 million people will suffer from hearing loss. The main Objective of the study was to determine the differences between the severity of the symptoms of stress, anxiety and depression in participants with varying degrees of sensorineural hearing loss during the COVID-19 pandemic. An additional aim was to examine the extent and manner in which protective face masks impact the communication of people with hearing loss. Matrials and Methods: A cross-sectional study was conducted, which included 160 patients (81 men and 79 women) with bilateral sensorineural hearing loss. The patients’ age range was 50 to 80 years. Depending on the degree of hearing loss or pure-tone threshold, the participants were divided into four groups: mild hearing loss, moderate hearing loss, severe hearing loss and profound hearing loss. The research used the Depression, Anxiety and Stress Scale (DASS-21) and a questionnaire in which the participants reported whether surgical face masks (medical three-layer masks) worn by speakers makes communication difficult, to what extent and in what way. Results: The average age of the patients was 67.97 ± 8.16. A significant correlation was found between the degree of hearing loss and communication difficulties caused by the use of protective face masks (*p* < 0.001). For patients with severe and profound hearing loss, communication is significantly more difficult (50.0% and 45.0% respectively) when the interlocutor wears a face mask. There is a significant correlation between the degree of hearing loss and the way in which communication is made more difficult when the interlocutor wears a face mask (*p* < 0.001). A statistically significant difference was determined between the degrees of hearing loss in all measured subscales: stress (*p* = 0.024), anxiety (*p* = 0.026) and depression (*p* = 0.016). Conclusions: We have determined that face masks used during the COVID-19 pandemic significantly hamper communication among the study groups (*p* = 0.007) and there is a significant correlation between the degree of sensorineural hearing loss and the presence of symptoms in all three DASS-21 subscales, meaning that the symptoms of stress, anxiety and depression were more intense in severe and profound hearing loss.

## 1. Introduction

The incidence of hearing loss is constantly increasing and, according to the World Health Organization (WHO), hearing loss had been determined in 466 million people worldwide until 2018. It has been estimated that by 2030, 630 million people worldwide will suffer from hearing loss and by 2050 the number will reach 900 million [1]. Modern research has shown that hearing loss can lead to some forms of mental disorder to occur with more frequency than in the general population, which are significantly contributed to by communication difficulties and social isolation [2,3]. Many studies have found a higher prevalence of depression [2,4], anxiety [5,6] and stress [7] in patients with sensorineural hearing loss, although there are studies where no statistically significant correlation was found between hearing loss and depression [8,9]. If mental disorders in people with sensorineural hearing loss are diagnosed in time, they may affect functional and cognitive ability as well as the quality of life of the patient [10].

The COVID-19 pandemic itself affects mental health and the development of the symptoms of stress, anxiety, depression, insomnia, anger and fear, all of which can even lead to suicide [11]. Fear and anxiety develop as adaptive reactions to uncertainty and impending danger which are present in the COVID-19 pandemic [12]. Protective measures such as face masks and physical distancing during the COVID-19 pandemic can further exacerbate the symptoms of stress, anxiety and depression in people with sensorineural hearing loss [13]. When the interlocutor is wearing a protective face mask, communication can be significantly more difficult for a person with impaired hearing because the interlocutor’s voice is less audible in certain frequencies [14,15]. It has been determined that a surgical mask reduces speech intelligibility by 23.3% especially in a noisy environment, while high filtration masks reduce speech intelligibility by as much as 69.0% [16]. For proper use of the protective masks, it is necessary to cover the mouth and the nose, i.e., 60–70% of the face, which makes it difficult to see the visible signs of the lips, facial expressions that help in speech perception, particularly when there is background noise [17] and especially in patients with impaired auditory function [18]. Speech audibility is also reduced due to the physical distance between the speakers the interlocutors, which can also cause an increase in background noise [16].

The main aim of the study was to determine the differences in the severity of the symptoms of stress, anxiety and depression in participants with varying degrees of sensorineural hearing loss during the COVID-19 pandemic.

The additional aim was to determine how the degree of hearing loss affected communication while people were wearing a face mask.

## 2. Materials and Methods

### 2.1. Study Design

The study was designed as a cross-sectional survey which included 160 patients (81 men and 79 women) with bilateral sensorineural hearing loss, aged 50 to 80 years. The study was conducted at the Otorhinolaryngology Clinic (ENT) of the University Clinical Center (UCC) Nis in the period from March to May 2021 during the COVID-19 pandemic.

Inclusion criteria were: bilateral sensorineural hearing loss of patients aged 50 to 80.

Exclusion criteria were: conductive and mixed hearing loss, congenital hearing anomalies, Meniere’s disease, posterior cranial fossa tumor, acoustic neurinoma, glomus tumor, tinnitus, incorrigible vision loss, psychiatric disorders, neurological disorders caused by alcohol and psychoactive substance abuse, patients with severe heart, kidney, lung and malignant diseases.

The research was approved by the Ethics Committee of the University Clinical Center Nis (Decision No. 2883, dated 28 January 2021).

### 2.2. Research Questionnaires

Data were collected using semi structured questionnaire specially created for this investigation. The questionnaire consisting of 18 questions and with two distinct sections.

Section 1: The first section explored the participants’ basic socio-demographic characteristics such as age and sex.

Section 2: The second section consisted of items divided into three subsections (the presence of comorbidities, alcohol and psychoactive substance abuse and the use of sign language and lip reading.

A Likert scale is a psychometric scale commonly used to represent participants opinions and attitudes to a topic or subject matter. The four point Likert scale was used in this study. Patients assessed each item on the Likert scale from zero-0 (it did not apply to me at all) to 3 (it concerns me most of the time) in relation to how often they felt that way during the last week.

The scores on each subscale multiplied by two, so the results ranged from 0 to 42, where higher scores indicate higher levels of anxiety, depression and stress.

The participants stated whether and to what extent (mild, moderate, severe) they experienced hearing difficulties in communication with an interlocutor wearing a surgical face mask (medical three-layer mask) as well as how communication is most impeded (I hear less sound, it is impossible to read from the lips, I do not understand what is said).

After signing an informed consent, the participants filled in the questionnaire with the help of an examiner.

Depression, Anxiety and Stress Scale (DASS-21) was used to assess the emotional state [19]. The DASS-21 is the short form of the DASS-42, a self-reported scale designed to measure the negative emotional states of depression, anxiety and stress. This scale is suitable for clinical settings to assist in diagnosis and outcome monitoring, as well as non-clinical settings as a mental health screener. This scale does not diagnose mental disorders but only assesses the frequency and intensity of symptoms.

The DASS-21 is based on a dimensional rather than a categorical conception of psychological disorders, and scores emphasize the degree to which someone is experiencing symptoms rather than having diagnostic cutoff points. DASS-21 is a standardized scale for self-assessment of three unpleasant emotional states and it contains three subscales each with seven items.

The first scale is about depression: dysphoria, hopelessness, devaluation of life, self-deprecation, lack of interest/involvement, anhedonia and inertia. (Items 3, 5, 10, 13, 16, 17, 21). The second scale is about anxiety: autonomic arousal, skeletal muscle effects, situational anxiety and subjective experience of anxious affect. (Items 2, 4, 7, 9, 15, 19, 20). The third scale is about stress: levels of chronic nonspecific arousal, difficulty relaxing, nervous arousal, and being easily upset/agitated, irritable/over-reactive and impatient. (Items 1, 6, 8, 11, 12, 14, 18). Score are presented as a total score and a score for the three subscales [20].

### 2.3. Diagnostic Examination

The diagnostic procedure included an audiological examination where the hearing threshold was determined by liminal tonal audiometry on a Diagnostic Audiometer AD 629 (Interacoustics, Middelfart, Denmark) in a soundproof room with pure tones at 128 Hz to 8 kHz frequencies. Hearing loss is defined by hearing the threshold in dB in the frequency range of 0.5 to 4 kHz. The degree of hearing loss was determined based on the assessment by the World Health Organization (WHO) where the following applied: 0–25 dB—normal hearing, 26–40 dB—mild hearing loss, 41–60 dB moderate, 61–80 dB severe, and >80 dB profound hearing loss [1]. In order to exclude conductive and mixed hearing loss, impedancemetry (tympanometry and stapedial reflex testing) was performed in addition to bone conduction. Depending on the degree of hearing loss or pure tone threshold, all participants were divided into four groups: mild hearing loss (Mild HL), moderate hearing loss (Mod HL), severe hearing loss (SHL) and profound hearing loss (PHL).

### 2.4. Statistical Analysis

The analysis of the obtained data was done using the SPSS 16.0 software package for Windows (SPSS Inc., Chicago, IL, USA). The obtained data are presented as arithmetic mean and standard deviation (SD). For the purposes of analysis, the data were also presented as absolute and relative numbers. The chi-squared test was used to analyze and test category features. The nonparametric Kruskal–Wallis H test was used to examine the differences in DASS-21 symptoms intensity among groups of patients with varying degrees of sensorineural hearing loss. The null hypothesis was tested with a significance threshold of *p* < 0.05.

## 3. Results

The study included 160 patients, average age 67.97 ± 8.16 years (min 50.0, max 80.0 years). The groups, which were formed in relation to the degree of hearing loss, were equalized according to gender (*p* = 0.965) and age (*p* = 0.491). Protective face masks used in the COVID-19 pandemic affected communication differently among the study groups and the differences were statistically significant (*p* = 0.007). The least affected was the Mild HL group (77.5%) whereas for the other groups, the difficulty was more even (90.9–97.5%). It was recorded that the patients’ ability to lip read differed statistically significantly between the examined groups (Table 1). Only two participants knew sign language and no statistical significance was determined with respect to the degree of hearing loss (*p* = 0.424).

Table 2 shows average values and standard deviation within each group, based on DASS-21 results. Among the groups, a statistically significant difference was found in all measured subscales: stress subscale (*p* = 0.024), anxiety subscale (*p* = 0.026) and depression subscale (*p* = 0.016).

A significant correlation was determined between the degree of hearing loss and difficulties in communication occurring due to the use of a protective face mask (*p* < 0.001). When the interlocutor wears a face mask, communication is significantly more difficult for patients with SHL and PHL (50.0% and 45.0%, respectively) (Table 3).

The degree of hearing loss is significantly related to the way in which communication is made more difficult due to mask wearing (*p* < 0.001). It was found that in addition to hearing less sound and not understanding what was said, a major subjective problem for patients with SHL and PHL was the inability to read from the lips, 32.5% to 37.5% (Table 4).

## 4. Discussion

The main aim of the study was to determine the differences in the severity of the symptoms of stress, anxiety and depression during the COVID-19 pandemic in participants with varying degrees of sensorineural hearing loss and to examine the extent to which and how communication is affected for people with hearing loss when the interlocutor uses a protective face mask. A sociodemographic questionnaire specially designed for the purposes of this study was used along with a questionnaire designed to examine the impacts of face masks on communication and a questionnaire for self-assessment of the symptoms of stress, depression and anxiety, DASS-21.

We found a significant relation between the degree of sensorineural hearing loss and the presence of the symptoms of stress, anxiety and depression where patients with SHL and PHL had higher scores on all three DASS-21 subscales, meaning that the symptoms of stress, anxiety and depression were the most severe in these two groups of participants. Many studies also suggest that impaired communication in people with sensorineural hearing loss may cause symptoms of stress, anxiety and depression [4,6,20,21]. Jaykody et al. examined older adult patients with hearing loss and also found a statistically significant association between the degree of hearing loss at speech frequencies and DASS-21 score highly on all three subscales [20].

A study by Blayer et al., showed that the symptoms of anxiety and depression are four times more common in people with hearing loss than in the general population [21]. In a meta-analysis by Lawrence et al., a significant prevalence of depression symptoms was found in older population with hearing loss [4]. Literature data indicate that the prevalence of clinically significant symptoms of anxiety is 15.4 to 31.3% in people with acquired hearing loss [5,6]. It has been found that untreated sensorineural hearing loss increases the risk of depression, anxiety and stress [22,23] making audiological diagnosis and auditory amplification necessary during the COVID-19 pandemic as well. Several studies suggest that the use of auditory amplifiers may reduce the symptoms of depression in patients [24,25].

Wearing a protective face mask reduces the audibility of the voice, prevents lip reading and the presentation of possible facial signals, which is particularly important in health facilities where coming to the facility itself is stressful for the patient and often there is background noise, which makes communication between the doctor and patient even more difficult [14,26]. These findings can explain the correlation that we found between the degree of hearing loss and communication difficulties caused by protective face masks. In our study, as well as in some others [18], patients with hearing loss frequently state that there is a major communication problem caused by the use of protective surgical masks because of the inability to read from the lips in addition to reduced audibility of the voice, which is particularly the case in severe and profound hearing loss. This can be explained by the fact that in communication, patients with sensorineural hearing loss rely more on visual signs than people with normal hearing [27].

The use of protective face masks increases the listening effort and impacts the certainty that the patient has understood correctly, which can cause misunderstanding during communication, increase stress and anxiety and lead to removal of the mask or reduction of the physical distance in order to improve communication [18,28].

Using DASS-21, Yang et al., found that compared to students with normal hearing, students with hearing loss during the COVID-19 pandemic had higher values on the stress subscale, which indicates the need for additional help and support in maintaining mental health and providing timely information on the pandemic [13].

The use of amplification technology (lapel microphone) may be one of the ways to improve verbal communication during the pandemic due to better auditory perception in people with and without hearing loss [15]. Studies also indicate that the use of transparent window masks provides better speech perception in people with hearing loss [14] but it should be noted that such masks are not widely accepted or available and they are acoustically unsuitable, i.e., sound distortion is greater than with surgical masks [29].

In the prevention of future pandemics, it is necessary to understand the importance of protective face masks and to find new options and strategies for fighting against this and other potential viral infections. In patients with hearing loss, when the speaker is wearing a mask, it is necessary to enable communication in such way that the speaker first attracts the listener’s attention, that the lighting is good, that they speak intelligibly and slowly and that they use gestures when they talk. It is also necessary to use hearing amplifiers (which can be adjusted to higher volume levels), cochlear implants, wireless systems and mobile phone applications [26]. It is required to always check the understanding of what has been said and to reduce the background noise. Written language can be used for important information and as an aid in communication and it is sometimes necessary to provide a sign language interpreter [30,31,32].

One of the limitations of our research is a small sample of respondents. Also, it does not take into account whether the participants or their families suffered from COVID-19, which may impact their mental status.

## 5. Conclusions

Based on the presented results, it can be concluded that there are differences in the severity of the symptoms of stress, anxiety and depression among participants with varying degrees of sensorineural hearing loss, where those with SHL and PHL reported more intense symptoms of stress, anxiety and depression. There is a correlation between the degree of hearing impairment of participants and disordered communication resulting from the use of a face masks during the COVID-19 pandemic.

## Figures and Tables

**Table 1 medicina-58-00233-t001:** Socio-demographic characteristics of patients in relation to the degree of hearing loss.

Variables	Mild HL	Mod HL	SHL	PHL	*p*
No. of cases	40	40	40	40	
Age (mean + SD) years	66.70 ± 7.17	67.65 ± 10.24	68.95 ± 7.44	68.58 ± 7.54	0.491 ^1^
Males/females *n*	19/21	21/19	21/19	20/20	0.965 ^2^
Protective mask affects communication *n* (%)	31 (77.5%)	36 (90.0%)	39 (97.5%)	39 (97.5%)	**0.007** ^2^
Knowledge of sign language *n* (%)	0 (0.0%)	1 (2.5%)	0 (0.0%)	1 (2.5%)	0.424 ^2^
Lip reading *n* (%)	2 (5.0%)	15 (37.5%)	11 (27.5%)	20 (50.0%)	**<0.001** ^2^

Mild HL: mild hearing loss, Mod HL: moderate hearing loss, SHL: severe hearing loss, PHL: profound hearing loss, *n*: number of subjects, SD: standard deviation, ^1^
*p*: Kruskal–Wallis H test value, ^2^
*p*: X^2^: chi-square test value (*p* < 0.05, bolded if significant).

**Table 2 medicina-58-00233-t002:** Differences in the severity of the symptoms of stress, anxiety and depression in groups with different degrees of hearing loss.

Variables	Mild HL	Mod HL	SHL	PHL	*p* ^1^
DASS stress, (mean + SD)	10.80 ± 6.79	13.70 ± 8.65	14.85 ± 10.15	17.55 ± 10.27	**0.024**
DASS anxiety, (mean + SD)	7.20 ± 5.69	8.50 ± 7.89	11.70 ± 10.66	13.70 ± 10.11	**0.026**
DASS depression (mean + SD)	7.15 ± 5.49	8.50 ± 7.55	11.15 ± 8.77	13.60 ± 10.09	**0.016**

DASS-21—Depression, Anxiety and Stress Scale, Mild HL: mild hearing loss, Mod HL: moderate hearing loss, SHL: severe hearing loss, PHL: profound hearing loss, SD: standard deviation, *p*
^1^: Kruskal–Wallis H test value (*p* < 0.05, bolded if significant).

**Table 3 medicina-58-00233-t003:** Effects of the degree of sensorineural hearing loss on communication due to the use of face mask.

Use of Face Mask	Mild HL *n* %		Mod HL *n* %		SHL *n* %		PHL *n* %		*p* ^2^
Does not make communication difficult	10	25.0	3	7.5	0	0.0	0	0.0	**<0.001**
Causes mild difficulties	12	30.0	8	20.0	7	17.5	2	5.0	
Causes moderate difficulties	13	32.5	17	42.5	13	32.5	20	50.0	
Causes significant difficulties	5	12.5	12	30.0	20	50.0	18	45.0	

Mild HL: mild hearing loss, Mod HL: moderate hearing loss, SHL: severe hearing loss, PHL: profound hearing loss, *n*: number of subjects, *p*
^2^: X^2^: chi-square test value (*p* < 0.05, bolded if significant).

**Table 4 medicina-58-00233-t004:** Relation between the degree of hearing loss and the type of communication difficulties caused by face masks.

Impact on Communication	Mild HL *n* %		Mod HL *n* %		SHL *n* %		PHL *n* %		*p* ^2^
No impact	9	22.5	4	10.0	0	0.0	0	0.0	**<0.001**
I hear less sound	15	37.5	13	32.5	11	27.5	12	30.0	
It is impossible to read from the lips	0	0.0	7	17.5	13	32.5	15	37.5	
I do not understand what was said	16	40.0	16	40.0	16	40.0	13	32.5	

Mild HL: mild hearing loss, Mod HL: moderate hearing loss, SHL: severe hearing loss, PHL: profound hearing loss, *n*: number of subjects, *p*
^2^: X^2^: chi-square test value (*p* < 0.05, bolded if significant).

## Data Availability

The data presented in this study are available on request from the corresponding author.

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
