# Peer review of "Impact of Sensorineural Hearing Loss during the Pandemic of COVID-19 on the Appearance of Depressive Symptoms, Anxiety and Stress"

_medicina, 2022, doi:10.3390/medicina58020233_

Round 1

Reviewer 1 Report

The manuscript deals with a very important issue, that is widely present in everyday clinical practice, hearing impairment and quality of life in people with hearing impairment. Sensorineural hearing loss leads to reduced comprehension of speech and communication disorders, and affects mental and emotional life as well as cognitive processes. In the Covid-19 pandemic, the use of a protective mask prevents reading from the lips, further reducing speech comprehension. The topic of the manuscript is interesting and current.Concept of the investigation is well designed. It is nicely presented.

Author Response

Response to reviewer 1 Commnets,

Very respected Reviewer,

We are very grateful for Your time to read our manuscript and for Your comments.

Reviewer 2 Report

The research is examining the  severity of the stress, anxiety and depression symptoms in participants with varying degrees of sensorineural hearing loss during the COVID-19 pandemic, and how masks  influence communication in this population . The theme is interesting to see how COVID-19 pandemic influenced this specific group of people. There are some remarks and suggestions:

Aim of the study

Authors should add one more aim- how the  degree of hearing loss impacted communication while wearing face mask.

Methods an material section

-Author should  change prospective study  to cross-section study, because  the study involved collecting  data from study population at one specific point in time. Prospective study would involve  following the population for a certain period of time, watching for outcomes (of the disease or a state) during the study period

-Authors should describe the used questioner in more detail. It consists of 18 questions, yet only socio-demographic characteristics of participants (sex and age) , mask effects on communication, the use of sign language and lip reading are shown in the table.

-Authors should describe DASS-21 scale in more detail, how many questions the scale contains, what are the subscales, range of values, what are the values indicating high levels of anxiety, depression and stress.

Discussion section

-The authors state the limitations of the study. First limitation can stand, but the second should be deleted. The title and the aim of the research was to determine if the symptoms of depression, anxiety and stress are present in population with hearing loss. Further assessment of these patient should be performed, but this isn’t a limitation (or the methodology of the study should have been differently formulated)

Conclusion section

The authors should only write conclusion statements which are direct result of the research. Broad statements about amplification,  team work in patients etc. with hearing loss should be  moved to the discussion section with appropriate  references.

Author Response

Response to Reviewer 2 Comments,

We are very grateful for Your  useful comments. 

Point 1. We changed the type of study as You suggested. Now it is a cross-sectional study.

Point 2. We added additioanl aim s You suggested. 

Point 3. We added more details in description of used general questionary and better explained how we used the four points Likert scale.

Point 4. We describe DASS-21 scale in details. We also added one more reference for this scale.

Point 5. We dilited the second limitation of the study as You suggested.

Point 6. We completed conclusion of the manuscript.